# Management preferences of orthopedic surgeons in rickets patients in Turkey: Results of a nationwide survey

Banu Turhan[1]*, Niyazi Erdem Yaşar[2], Mehmet Ali Sungur[3], Yalçın Turhan[2], Şenol Bekmez[2]

**1** Department of Pediatric Endocrinology, Ankara Atatürk Sanatoryum Education and Research Hospital, Ankara, Turkey, **2** Department of Orthopedics and Traumatology, Health Sciences University, Bilkent Children's Hospital, Ankara, Turkey, **3** Department of Biostatistics and Medical Informatics, Duzce Medical Faculty, Duzce, Turkey

☉ All authors contributed equally to this work.
* benguulkuturhan@yahoo.com

## Abstract

### Aim

Rickets remains a significant health issue, particularly in developing countries. This study evaluated the management preferences of orthopedic surgeons in Turkey regarding rickets and analyzed these approaches from a pediatric endocrinology perspective.

### Methods

An online survey was developed based on a comprehensive literature review and previous similar studies. The survey link was distributed via email to members of the Turkish Orthopedic and Traumatology Association and the Pediatric Orthopedics Association. The questionnaire included 14 multiple-choice items addressing socio-demographic characteristics and the general approach to the diagnosis and treatment of rickets. Prior to dissemination, the survey was reviewed by 13 board-certified orthopedic surgeons from various institutions to ensure clarity and content validity.

### Results

Of the 257 respondents, 198 met the inclusion criteria and were included in the analysis. Orthopedic surgeons with more than 10 years of experience (n = 111,56.1%) were significantly less likely to refer rickets patients to a pediatric endocrinologist (p = 0.009) and more likely to recommend oral or intramuscular 25-OH-vitamin D3 ampoule treatment rather than oral drops or capsules (p < 0.001). Surgeons who saw fewer than 25% pediatric patients in daily practice (n = 110,55.6%) were more inclined to refer patients without ordering diagnostic tests or initiating treatment (p = 0.011).

**Data availability statement:** All relevant data are within the paper and its Supporting Information files.

**Funding:** The author(s) received no specific funding for this work.

**Competing interests:** The authors have declared that no competing interests exist.

The presence of a pediatric endocrinologist within the same institution (n = 96,48.5%) was also associated with increased referral rates without preliminary testing or treatment ($p = 0.003$/$p = 0.002$, respectively). Physicians who encountered at least one rickets patient per week (n = 78,39.4%) demonstrated better knowledge of normal serum 25-OH-vitamin D3 levels ($p = 0.042$) and were significantly less likely to refer patients to a pediatric endocrinologist ($p < 0.001$).

## Conclusion

Management approaches to rickets among orthopedic and traumatology specialists in Turkey vary significantly depending on clinical experience, practice setting, and access to a pediatric endocrinologist. To prevent both undertreatment and unnecessary referrals, newly diagnosed rickets patients in orthopedic clinics should be appropriately evaluated and referred when necessary. Establishing a standardized diagnostic and referral algorithm based on international guidelines and incorporating it into orthopedic training programs is strongly recommended.

## Introduction

Rickets is characterized by inadequate mineralization of the growth plates and bone matrix of the skeleton and represents a heterogeneous group of disorders with both acquired and inherited subtypes. Patients with rickets typically present with decreased serum calcium and/or phosphorus levels [1]. In addition to various skeletal deformities, hypocalcemia-related symptoms, such as tetany, convulsions, laryngospasm, and muscle weakness, may also be observed [2]. Although nutritional rickets, caused by insufficient intake of vitamin D or calcium, is the most commonly encountered subtype, recognition of inherited forms, including hypophosphatemic and vitamin D-dependent rickets, is increasing due to advances in molecular genetic diagnostic techniques [3].

Despite the implementation of prophylactic treatment protocols in many countries to prevent rickets, the disease remains a significant public health issue, particularly in developing nations. In 2005, a marked decline in the incidence of rickets was reported in Turkey following the introduction of the "Vitamin D Deficiency Prevention Program" [4,5]. However, recent studies have indicated a resurgence in rickets cases, particularly due to inappropriate or inadequate administration of vitamin D supplementation intended for prophylaxis [6,7]. One major contributing factor to this increase is the insufficient follow-up and treatment of the children of refugees who have migrated to the country for various reasons. Similar challenges have been reported in several other countries as well [8,9].

Although physical examination, medical history, and biochemical analyses provide the primary basis for diagnosing rickets, radiological confirmation is essential [1]. Physical examination alone is insufficient to differentiate between children with active rickets and those in recovery, and it does not allow for distinction between the various subtypes of the disease [10]. Moreover, if skeletal abnormalities are detectable on

clinical examination, the disease is often already at an advanced stage. Therefore, reliance solely on clinical signs may lead to overestimation of the prevalence of active rickets and result in unnecessary treatment.

Rickets, which holds significant relevance in orthopedic and traumatology practice, often presents through various skeletal deformities and pathological fractures. Patients with nutritional rickets, in particular, may be evaluated by orthopedic surgeons at any stage of the disease course [11]. The most common reasons for referral to orthopedic and traumatology departments include nonspecific musculoskeletal pain, genu varum, genu valgum, and fractures that occur spontaneously or after minor trauma [12–14].

Rickets is a serious pediatric health concern, and early diagnosis and treatment are crucial to prevent the development of permanent and significant sequelae later in life. Given that different subtypes of rickets require tailored treatment approaches, establishment of an accurate and standardized treatment protocol is essential [15]. This study evaluated the clinical approaches of orthopedic surgeons who frequently encounter patients with rickets and who play pivotal roles in both diagnosis and disease management. The findings of this study may contribute to the development of more accurate and standardized diagnostic and therapeutic strategies for rickets, fostering better collaboration between pediatric endocrinologists and orthopedic surgeons.

## Materials and methods

Following a comprehensive review of the literature and previous related studies, an online questionnaire was developed and distributed to orthopedic surgeons who are members of the Turkish Society of Orthopedics and Traumatology and the Pediatric Orthopedics Association. Prior to questionnaire development, a detailed MEDLINE search was conducted, revealing that no similar study had targeted orthopedic surgeons previously. Ethical approval for the study was obtained from the Ankara Bilkent City Hospital Medical Research Ethics Committee (dated February 14, 2024, number 1-24-15). The survey link, hosted on https://docs.google.com/forms, was distributed via email. As of 1 October 2024, the survey link started to be sent to the participants, three reminder e-mails were sent at 3-week intervals to encourage participation, and the survey link activation was terminated on 30 November 2024. With the online link sent to the participating physicians, before starting the survey the detailed information about the purpose and content of the study was given and their online informed consent was obtained.

The questionnaire comprised 14 multiple-choice and open-ended textbox questions designed to assess participants' sociodemographic characteristics (including institution of employment, years of professional experience, frequency of treating pediatric and rickets patients, and the presence of a pediatric endocrinologist at their institution) as well as their general approach to the diagnosis and management of rickets (S1 Appendix). Prior to distribution, the questionnaire was reviewed by 13 board-certified orthopedic surgeons from various institutions to evaluate its clarity and identify potential issues. Necessary revisions were made based on their feedback.

The first three questions gathered demographic information about the participants. Subsequent questions addressed whether a pediatric endocrinologist was available at each participant's institution and the frequency with which patients with rickets were encountered in daily practice. Later sections of the questionnaire were designed to evaluate participants' clinical approaches to suspected rickets cases, including the clinical and radiological findings they anticipated, concerns typically expressed by parents, laboratory tests ordered, treatment strategies employed, and their awareness of potential side effects related to the treatments.

Physicians who did not treat patients aged 0–18 years were instructed not to proceed with the survey. Upon collecting the responses, data analysis was limited to a specific subset of participants, i.e., those who were orthopedic specialists, who regularly treated pediatric patients within the 0–18 age group, and who had completed all survey questions. Responses from individuals who did not meet these criteria, including those who were not orthopedic specialists, who did not treat pediatric patients, or who submitted incomplete responses, were excluded from the final evaluation.

The resulting data were analyzed in detail according to four main parameters: the number of years the respondent had worked as an orthopedic specialist, the frequency of encountering pediatric patients in daily practice, the presence or absence of a pediatric endocrinologist at their institution, and the frequency of encountering patients with rickets.

## Statistical analysis

Statistical analyses were performed using IBM SPSS Statistics for Windows, version 22.0 (IBM Corp., Released 2013, Armonk, NY, USA). Qualitative variables were presented as numbers and percentages. The Pearson chi-square test, Fisher's exact test, or the Fisher-Freeman-Halton test was used, depending on the expected cell counts and the number of categories being compared. A *p*-value of < 0.05 was considered statistically significant.

## Results

Among the 257 completed surveys, 198 met the inclusion criteria. Approximately half of the participants had more than 10 years of experience as orthopedic surgeons (n = 111, 56.1%), managed pediatric patients in ≥ 25% of their daily practice (n = 88, 44.4%), and encountered at least one rickets patient per week (n = 78, 39.4%). Nearly half of the respondents reported the presence of a pediatric endocrinologist at their institution (n = 96, 48.5%). Additionally, the majority of participants (n = 139, 70.2%) were affiliated with government healthcare facilities (Table 1).

Participants' responses were analyzed according to years of experience in orthopedic surgery (Table 2). As years of experience increased, the likelihood of requesting serum calcium and 25-hydroxyvitamin D3 (25-OH-vitamin D3) for a diagnosis of rickets significantly increased (*p* = 0.008). When asked about radiological findings expected in pediatric rickets

**Table 1. Demographic data of participants.**

|  | n (%) |
|---|---|
| **Experience in years as orthopedic surgeon** |  |
| <5 years | 73 (36.9) |
| 5-10 years | 38 (19.2) |
| 11-15 years | 27 (13.6) |
| 16-20 years | 37 (18.7) |
| >20 years | 23 (11.6) |
| **Rate of seeing pediatric patients in daily practice** |  |
| <25% | 110 (55.6) |
| 25-50% | 27 (13.6) |
| 51-75% | 34 (17.2) |
| >75% | 27 (13.6) |
| **Institution** |  |
| Government Hospital | 38 (19.2) |
| Education Research/City Hospital | 69 (34.8) |
| Government University Hospital | 32 (16.2) |
| Foundation/Private University Hospital | 20 (10.1) |
| Private Hospital | 19 (9.6) |
| Private Practice | 20 (10.1) |
| **Is there a staff pediatric endocrinologist in your institute?** |  |
| Present | 96 (48.5) |
| Absent | 102 (51.5) |
| **Frequency of rickets patient presentation** |  |
| Every day | 14 (7.1) |
| >1 in a week | 42 (21.2) |
| 1 in a week | 22 (11.1) |
| 1 in a month | 37 (18.7) |
| <1 in a month | 83 (41.9) |

**Table 2. Comparison according to the experience in years as an orthopedic surgeon.**

| Question | ≤10years (n=111) | >10years (n=87) | p |
|---|---|---|---|
| **What are the most common complaints of parents suggesting that their child has rickets?** | | | |
| Enlargement of the wrists and ankles | 11 (9.9) | 16 (18.4) | 0.084 |
| Leg and knee deformities (genu varum. genu valgum) | 104 (93.7) | 84 (96.6) | 0.517 |
| Enlargement of the costochondral junction (rachitic rosary) | 10 (9.0) | 27 (31.0) | **<0.001** |
| Thoracal deformities | 10 (9.0) | 14 (16.1) | 0.130 |
| **What are the concerns that families convey to you about rickets?** | | | |
| Gait disorder | 53 (47.7) | 37 (42.5) | 0.464 |
| Decreased growth in height | 23 (20.7) | 34 (39.1) | **0.005** |
| Having bone fractures | 15 (13.5) | 41 (47.1) | **<0.001** |
| Deformity in bones | 76 (68.5) | 78 (89.7) | **<0.001** |
| **What tests would you order for a child who is suspected of having rickets?** | | | |
| I'll do a referral to a pediatric endocrinologist without a test order. | 50 (45.0) | 28 (32.2) | 0.066 |
| Calcium | 49 (44.1) | 55 (63.2) | **0.008** |
| Phosphorus | 45 (40.5) | 32 (36.8) | 0.590 |
| Magnesium | 24 (21.6) | 13 (14.9) | 0.231 |
| Alkaline phosphatase | 43 (38.7) | 34 (39.1) | 0.961 |
| 25-OH-vit-D3 | 53 (47.7) | 58 (66.7) | **0.008** |
| Skeletal survey | 18 (16.2) | 10 (11.5) | 0.344 |
| Bone X-rays | 65 (58.6) | 57 (65.5) | 0.318 |
| **What kind of radiological findings would you expect to see in a child with a suspected case of rickets?** | | | |
| Metaphyseal enlargement, irregularity and clubbing | 100 (90.1) | 85 (97.7) | **0.032** |
| Growth plate enlargement | 62 (55.9) | 71 (81.6) | **<0.001** |
| Osteopenia in the long bones | 49 (44.1) | 48 (55.2) | 0.123 |
| Stress fractures | 37 (33.3) | 50 (57.5) | **0.001** |
| Pelvic deformities | 22 (19.8) | 38 (43.7) | **<0.001** |
| **Which of the following are types of rickets?** | | | |
| Nutritional (calciopenic, phosphopenic) | 92 (82.9) | 81 (93.1) | **0.032** |
| Vitamin D-resistant (familial hypophosphatemic) | 96 (86.5) | 83 (95.4) | **0.035** |
| Vitamin D-dependent (genetic) | 84 (75.7) | 74 (85.1) | 0.103 |
| **Which option specifies the normal value of 25 OH vitD3?** | | | |
| <12 ng/ml <30 nmol/L | 4 (3.6) | 0 (0.0) | 0.132 |
| 12-20 ng/ml 30–50 nmol/L | 33 (29.7) | 30 (34.5) | 0.476 |
| >20 ng/ml >50 nmol/L | 66 (59.5) | 59 (67.8) | 0.226 |
| >100 ng/ml >250 nmol/L | 8 (7.2) | 2 (2.3) | 0.190 |
| **For your patients presenting with complaints/findings of rickets, would you question whether they are regularly taking the vitamin D treatment recommended by their Family Physicians under the age of 1 year?** | | | |
| Yes | 82 (73.9) | 77 (88.5) | **0.010** |
| **In patients with 25 OH-VitD3 deficiency, how would you proceed with conservative treatment?** | | | |
| I do not start treatment; I recommend consulting a pediatric endocrinologist | 81 (73.0) | 48 (55.2) | **0.009** |
| I recommend a diet rich in calcium and phosphorus | 23 (20.7) | 40 (46.0) | **<0.001** |
| I start oral 25 OH vitD3 drops | 30 (27.0) | 18 (20.7) | 0.302 |
| I start oral 25 OH vitD3 ampoules | 4 (3.6) | 27 (31.0) | **<0.001** |
| I start intramuscular 25 OH vitD3 ampoules | 2 (1.8) | 19 (21.8) | **<0.001** |
| I start oral 25 OH vitD3 tablets/capsules | 3 (2.7) | 12 (13.8) | **0.003** |
| I recommend increasing the frequency of sunbathing | 37 (33.3) | 51 (58.6) | **<0.001** |

*(Continued)*

**Table 2.** (Continued)

| Question | ≤10years (n = 111) | >10years (n = 87) | p |
|---|---|---|---|
| **What side-effect(s) are associated with vitamin D preparations if used in an uncontrolled way?** | | | |
| Weakness. fatigue | 55 (49.5) | 49 (56.3) | 0.344 |
| Anorexia | 25 (22.5) | 25 (28.7) | 0.318 |
| Bone pain | 44 (39.6) | 28 (32.2) | 0.279 |
| Kidney stone formation | 92 (82.9) | 74 (85.1) | 0.680 |
| Cardiac arrhythmia | 50 (45.0) | 28 (32.2) | 0.066 |
| Pancreatitis | 46 (41.4) | 18 (20.7) | **0.002** |
| Coma | 16 (14.4) | 6 (6.9) | 0.095 |

cases, participants with more than 10 years of experience more frequently reported metaphyseal and growth plate widening, stress fractures, and pelvic deformities as expected findings ($p = 0.032, < 0.001, = 0.001$, and $< 0.001$, respectively). While the majority of participants selected $> 20$ ng/mL ($> 50$ nmol/L) as the normal 25-OH-vitamin D3 level, no significant difference was observed between experience groups. Experienced surgeons were significantly less likely to refer patients to a pediatric endocrinologist ($p = 0.009$) and more likely to recommend oral or intramuscular 25-OH-vitamin D3 ampoule treatments ($p < 0.001$).

When participants were grouped based on the proportion of pediatric patients seen in daily practice, those who treated $< 25\%$ pediatric patients were more likely to refer rickets cases to a pediatric endocrinologist without ordering diagnostic tests ($p = 0.011$). As the proportion of pediatric patients increased, significant differences emerged regarding the radiological findings reported for rickets cases. Moreover, surgeons treating a higher proportion of pediatric patients more frequently identified $> 20$ ng/mL ($> 50$ nmol/L) as the normal value for 25-OH-vitamin D3 ($p = 0.027$). Those treating $< 25\%$ pediatric patients were more inclined to refer rickets patients to pediatric endocrinology without initiating treatment ($p = 0.005$) (Table 3).

Surgeons working in institutions with a pediatric endocrinologist were significantly more likely to refer suspected rickets cases directly to the endocrinologist without ordering laboratory tests ($p = 0.003$) or initiating treatment ($p = 0.002$). These surgeons were also less likely to prescribe oral or intramuscular 25-OH-vitamin D3 ampoules ($p < 0.001$) (Table 4).

Participants who saw $\geq 1$ rickets patient per week demonstrated significantly different response patterns concerning expected radiological findings ($p = 0.016$ and $< 0.001$). These surgeons also more frequently identified $> 20$ ng/mL ($> 50$ nmol/L) as the normal 25-OH-vitamin D3 level ($p = 0.042$). Conversely, surgeons who encountered $< 1$ rickets patient per week were more likely to refer these patients to a pediatric endocrinologist rather than initiate treatment themselves ($p < 0.001$), and were less likely to prescribe oral or intramuscular 25-OH-vitamin D3 ($p < 0.001$) (Table 5).

Finally, when asked whether they would inquire whether a patient suspected of having rickets had received vitamin D supplementation from a family physician during infancy ($< 1$ year of age), significantly more affirmative responses were given by physicians with $> 10$ years of experience ($p = 0.010$), those treating $\geq 25\%$ pediatric patients ($p = 0.008$), and those seeing $\geq 1$ rickets patient per week ($p = 0.020$) (Tables 2–3, and 5).

## Discussion

This study used an electronic questionnaire to evaluate the clinical approaches of orthopedic surgeons to patients with suspected rickets. Physicians with more than 10 years of professional experience were less likely to refer such patients to a pediatric endocrinologist and more likely to initiate treatment with oral or intramuscular 25-OH-vitamin D3 ampoules. Conversely, those who managed fewer pediatric patients ($< 25\%$ of their caseload) and those working in clinics with an on-site pediatric endocrinologist were more inclined to refer patients to the endocrinologist without conducting further investigations.

**Table 3. Comparison according to the rate of seeing pediatric patients in daily practice.**

| Question | <25% (n = 110) | ≥25% (n = 88) | p |
|---|---|---|---|
| **What are the most common complaints of parents suggesting that their child has rickets?** | | | |
| Enlargement of the wrists and ankles | 8 (7.3) | 19 (21.6) | **0.004** |
| Leg and knee deformities (genu varum, genu valgum) | 101 (91.8) | 87 (98.9) | **0.045** |
| Enlargement of the costochondral junction (rachitic rosary) | 11 (10.0) | 26 (29.5) | **<0.001** |
| Thoracal deformities | 9 (8.2) | 15 (17.0) | 0.058 |
| **What are the concerns that families convey to you about rickets?** | | | |
| Gait disorder | 52 (47.3) | 38 (43.2) | 0.566 |
| Decreased growth in height | 26 (23.6) | 31 (35.2) | 0.073 |
| Having bone fractures | 11 (10.0) | 45 (51.1) | **<0.001** |
| Deformity in bones | 74 (67.3) | 80 (90.9) | **<0.001** |
| **What tests would you order for a child who is suspected of having rickets?** | | | |
| I'll do a referral to a pediatric endocrinologist without a test order. | 52 (47.3) | 26 (29.5) | **0.011** |
| Calcium | 48 (43.6) | 56 (63.6) | **0.005** |
| Phosphorus | 45 (40.9) | 32 (36.4) | 0.514 |
| Magnesium | 20 (18.2) | 17 (19.3) | 0.838 |
| Alkaline phosphatase | 40 (36.4) | 37 (42.0) | 0.415 |
| 25-OH-vit-D3 | 50 (45.5) | 61 (69.3) | **0.001** |
| Skeletal survey | 13 (11.8) | 15 (17.0) | 0.294 |
| Bone X-rays | 59 (53.6) | 63 (71.6) | **0.010** |
| **What kind of radiological findings would you expect to see in a child with a suspected case of rickets?** | | | |
| Metaphyseal enlargement, irregularity and clubbing | 99 (90.0) | 86 (97.7) | **0.029** |
| Growth plate enlargement | 56 (50.9) | 77 (87.5) | **<0.001** |
| Osteopenia in the long bones | 37 (33.6) | 60 (68.2) | **<0.001** |
| Stress fractures | 30 (27.3) | 57 (64.8) | **<0.001** |
| Pelvic deformities | 13 (11.8) | 47 (53.4) | **<0.001** |
| **Which of the following are types of rickets?** | | | |
| Nutritional (calciopenic, phosphopenic) | 90 (81.8) | 83 (94.3) | **0.009** |
| Vitamin D-resistant (familial hypophosphatemic) | 92 (83.6) | 87 (98.9) | **<0.001** |
| Vitamin D-dependent (genetic) | 80 (72.7) | 78 (88.6) | **0.006** |
| **Which option specifies the normal value of 25 OH vitD3?** | | | |
| <12 ng/ml <30 nmol/L | 4 (3.6) | 0 (0.0) | 0.130 |
| 12-20 ng/ml 30–50 nmol/L | 38 (34.5) | 25 (28.4) | 0.357 |
| >20 ng/ml >50 nmol/L | 62 (56.4) | 63 (71.6) | **0.027** |
| >100 ng/ml >250 nmol/L | 7 (6.4) | 3 (3.4) | 0.517 |
| **For your patients presenting with complaints/findings of rickets, would you question whether they are regularly taking the vitamin D treatment recommended by their Family Physicians under the age of 1 year?** | | | |
| Yes | 81 (73.6) | 78 (88.6) | **0.008** |
| **In patients with 25 OH-VitD3 deficiency, how would you proceed with conservative treatment?** | | | |
| I do not start treatment; I recommend consulting a pediatric endocrinologist | 81 (73.6) | 48 (54.5) | **0.005** |
| I recommend a diet rich in calcium and phosphorus | 19 (17.3) | 44 (50.0) | **<0.001** |
| I start oral 25 OH vitD3 drops | 33 (30.0) | 15 (17.0) | **0.035** |
| I start oral 25 OH vitD3 ampoules | 5 (4.5) | 26 (29.5) | **<0.001** |
| I start intramuscular 25 OH vitD3 ampoules | 3 (2.7) | 18 (20.5) | **<0.001** |
| I start oral 25 OH vitD3 tablets/capsules | 4 (3.6) | 11 (12.5) | **0.019** |
| I recommend increasing the frequency of sunbathing | 34 (30.9) | 54 (61.4) | **<0.001** |

*(Continued)*

**Table 3.** (Continued)

| Question | <25% (n = 110) | ≥25% (n = 88) | p |
|---|---|---|---|
| **What side-effect(s) are associated with vitamin D preparations if used in an uncontrolled way?** | | | |
| Weakness. fatigue | 54 (49.1) | 50 (56.8) | 0.279 |
| Anorexia | 28 (25.5) | 22 (25.0) | 0.942 |
| Bone pain | 48 (43.6) | 24 (27.3) | **0.017** |
| Kidney stone formation | 85 (77.3) | 81 (92.0) | **0.005** |
| Cardiac arrhythmia | 50 (45.5) | 28 (31.8) | 0.051 |
| Pancreatitis | 40 (36.4) | 24 (27.3) | 0.174 |
| Coma | 12 (10.9) | 10 (11.4) | 0.919 |

It is essential that children with suspected rickets are evaluated by a pediatric endocrinologist to ensure accurate diagnosis and standardized treatment, as rickets remains a significant health issue in many developing countries and includes various subtypes [16]. Orthopedic manifestations of rickets in early childhood can often be resolved with timely and appropriate replacement therapy, or at the very least, optimal pre-treatment can improve surgical outcomes [11]. Thus, effective collaboration between orthopedic surgeons and pediatric endocrinologists is crucial in managing rickets patients.

Diagnosis of rickets relies on a comprehensive evaluation including physical examination, family history, laboratory analyses, and radiological findings. The most frequently observed clinical signs include wrist and ankle widening, genu varum or valgum, thoracic deformities, and prominent costochondral junctions [17]. In this study, while physicians were asked to report common clinical complaints and reasons for hospital visits as noted by families, no significant differences were observed based on their professional experience or the frequency with which they managed pediatric or rickets cases.

For accurate diagnosis and classification of rickets subtypes, laboratory tests such as serum calcium, phosphate, magnesium, alkaline phosphatase, and 25-OH-vitamin D3 levels are essential [1]. Although our study did not find significant differences in test-ordering behavior among physicians overall, those treating fewer pediatric patients were significantly more likely to refer patients to a pediatric endocrinologist without ordering any laboratory tests ($p = 0.011$).

Radiological evaluation plays a crucial role in diagnosing rickets. Common findings include metaphyseal and growth plate widening, osteopenia of the long bones, stress fractures, and pelvic deformities [18]. Our findings reveal that surgeons who saw ≥ 25% pediatric patients or ≥ 1 rickets patient per week had significantly greater awareness of these radiographic features ($p < 0.001$), implying that frequent exposure to such cases enhances diagnostic confidence and knowledge.

Although there is some variability in the literature regarding the cutoff for normal 25-OH-vitamin D3 levels, values above 50 nmol/L (> 20 ng/mL) are generally accepted as normal [11]. In our study, while no statistically significant difference was found overall, surgeons who regularly treated pediatric patients or managed ≥ 1 rickets case per week identified the correct threshold value more frequently ($p = 0.027$ and = 0.042, respectively).

Since the implementation of Turkey's "Vitamin D Prophylaxis Augmentation Programme" in 2005, there has been a marked decrease in the incidence of rickets [5]. However, due to recent demographic changes, particularly an influx of regular and irregular migrants, monitoring and maintaining optimal vitamin D levels in early childhood have become increasingly challenging, potentially leading to a resurgence of nutritional rickets as a public-health concern [7–9]. Family physicians play a key role in preventing vitamin D deficiency by providing routine oral 25-OH-vitamin D3 supplementation to children under the age of 1 year. In our study, physicians with > 10 years of experience, those treating ≥ 25% pediatric patients, and those who managed ≥ 1 rickets case per week were significantly more likely to inquire about prior vitamin D

**Table 4. Comparison according to whether there is a pediatric endocrinologist working in the institution.**

| Question | Present (n = 96) | Absent (n = 102) | p |
|---|---|---|---|
| **What are the most common complaints of parents suggesting that their child has rickets?** | | | |
| Enlargement of the wrists and ankles | 17 (17.7) | 10 (9.8) | 0.105 |
| Leg and knee deformities (genu varum, genu valgum) | 87 (90.6) | 101 (99.0) | **0.008** |
| Enlargement of the costochondral junction (rachitic rosary) | 19 (19.8) | 18 (17.6) | 0.699 |
| Thoracal deformities | 15 (15.6) | 9 (8.8) | 0.143 |
| **What are the concerns that families convey to you about rickets?** | | | |
| Gait disorder | 43 (44.8) | 47 (46.1) | 0.856 |
| Decreased growth in height | 28 (29.2) | 29 (28.4) | 0.909 |
| Having bone fractures | 21 (21.9) | 35 (34.3) | 0.052 |
| Deformity in bones | 72 (75.0) | 82 (80.4) | 0.362 |
| **What tests would you order for a child who is suspected of having rickets?** | | | |
| I'll do a referral to a pediatric endocrinologist without a test order. | 48 (50.0) | 30 (29.4) | **0.003** |
| Calcium | 45 (46.9) | 59 (57.8) | 0.122 |
| Phosphorus | 43 (44.8) | 34 (33.3) | 0.098 |
| Magnesium | 20 (20.8) | 17 (16.7) | 0.452 |
| Alkaline phosphatase | 39 (40.6) | 38 (37.3) | 0.627 |
| 25-OH-vit-D3 | 48 (50.0) | 63 (61.8) | 0.096 |
| Skeletal survey | 14 (14.6) | 14 (13.7) | 0.863 |
| Bone X-rays | 57 (59.4) | 65 (63.7) | 0.529 |
| **What kind of radiological findings would you expect to see in a child with a suspected case of rickets?** | | | |
| Metaphyseal enlargement, irregularity and clubbing | 92 (95.8) | 93 (91.2) | 0.186 |
| Growth plate enlargement | 61 (63.5) | 72 (70.6) | 0.291 |
| Osteopenia in the long bones | 45 (46.9) | 52 (51.0) | 0.564 |
| Stress fractures | 41 (42.7) | 46 (45.1) | 0.735 |
| Pelvic deformities | 24 (25.0) | 36 (35.3) | 0.115 |
| **Which of the following are types of rickets?** | | | |
| Nutritional (calciopenic, phosphopenic) | 80 (83.3) | 93 (91.2) | 0.097 |
| Vitamin D-resistant (familial hypophosphatemic) | 87 (90.6) | 92 (90.2) | 0.918 |
| Vitamin D-dependent (genetic) | 74 (77.1) | 84 (82.4) | 0.356 |
| **Which option specifies the normal value of 25 OH vitD3?** | 3 (3.1) | 1 (1.0) | 0.357 |
| <12 ng/ml <30 nmol/L | 29 (30.2) | 34 (33.3) | 0.637 |
| 12-20 ng/ml 30–50 nmol/L | 60 (62.5) | 65 (63.7) | 0.858 |
| >20 ng/ml >50 nmol/L | 6 (6.3) | 4 (3.9) | 0.528 |
| >100 ng/ml >250 nmol/L | | | |
| **For your patients presenting with complaints/findings of rickets, would you question whether they are regularly taking the vitamin D treatment recommended by their Family Physicians under the age of 1 year?** | | | |
| Yes | 73 (76.0) | 86 (84.3) | 0.144 |
| **In patients with 25 OH-VitD3 deficiency, how would you proceed with conservative treatment?** | | | |
| I do not start treatment; I recommend consulting a pediatric endocrinologist | 73 (76.0) | 56 (54.9) | **0.002** |
| I recommend a diet rich in calcium and phosphorus | 26 (27.1) | 37 (36.3) | 0.165 |
| I start oral 25 OH vitD3 drops | 26 (27.1) | 22 (21.6) | 0.365 |
| I start oral 25 OH vitD3 ampoules | 6 (6.3) | 25 (24.5) | **<0.001** |
| I start intramuscular 25 OH vitD3 ampoules | 2 (2.1) | 19 (18.6) | **<0.001** |
| I start oral 25 OH vitD3 tablets/capsules | 3 (3.1) | 12 (11.8) | **0.022** |
| I recommend increasing the frequency of sunbathing | 38 (39.6) | 50 (49.0) | 0.182 |

*(Continued)*

**Table 4.** (Continued)

| Question | Present (n = 96) | Absent (n = 102) | p |
|---|---|---|---|
| **What side-effect(s) are associated with vitamin D preparations if used in an uncontrolled way?** | | | |
| Weakness. fatigue | 53 (55.2) | 51 (50.0) | 0.463 |
| Anorexia | 25 (26.0) | 25 (24.5) | 0.804 |
| Bone pain | 34 (35.4) | 38 (37.3) | 0.788 |
| Kidney stone formation | 79 (82.3) | 87 (85.3) | 0.566 |
| Cardiac arrhythmia | 49 (51.0) | 29 (28.4) | **0.001** |
| Pancreatitis | 32 (33.2) | 32 (31.4) | 0.768 |
| Coma | 16 (16.7) | 6 (5.9) | **0.016** |

supplementation in suspected cases ($p = 0.010$, = 0.008, and = 0.020, respectively), reflecting a higher level of clinical vigilance in these subgroups.

Treatment modalities for rickets due to nutritional deficiencies of vitamin D, calcium, or phosphorus vary significantly depending on the patient's age and the underlying etiology [2,19]. In cases of rickets caused by vitamin D deficiency, treatment typically involves oral or intramuscular administration of 25-OH-vitamin D3, and calcium supplementation may be required based on the individual patient's needs [20]. In contrast, calcium-deficiency rickets is primarily managed with calcium replacement, while vitamin D supplementation may be added to facilitate rapid symptomatic improvement in certain cases [21]. For rickets due to nutritional phosphorus deficiency, it is crucial to assess parathyroid hormone levels before initiating treatment. Parathyroid hormone levels are typically elevated in calcium deficiency; however, in phosphorus deficiency, they may be decreased, normal, or elevated, depending on the specific pathophysiological mechanisms involved [1].

Vitamin D-dependent and hypophosphatemic rickets follow different therapeutic protocols compared to nutritional rickets and generally require more complex individualized management strategies [22,23]. As such, a comprehensive clinical evaluation is essential in patients with suspected rickets to guide the development of patient-specific treatment plans.

In our study, we observed that orthopedic surgeons with more than 10 years of professional experience and those who managed pediatric patients in more than 25% of their daily clinical practice were significantly more likely to both diagnose and treat rickets independently ($p = 0.009$ and = 0.005, respectively). Furthermore, this subgroup demonstrated a greater tendency to recommend oral 25-OH-vitamin D3 ampoule therapy ($p < 0.001$). However, in our country, the only available formulation of 25-OH-vitamin D3 in ampoule form contains 300,000 IU/mL, which raises concerns regarding accurate dose adjustment and patient compliance, rendering this formulation suboptimal for general use [24].

Although vitamin D deficiency is associated with severe health consequences, inappropriate or unsupervised treatment can lead to equally significant complications. While some adverse effects of excessive vitamin D are mild and manageable, others may result in serious and irreversible outcomes [24]. Toxic doses of vitamin D may induce complications such as nephrolithiasis, cardiac arrhythmias, pancreatitis, and even coma, in addition to more common symptoms such as weakness, fatigue, anorexia, and bone pain. Moreover, in patients with underlying genetic mutations affecting enzymes like vitamin D hydroxylase, severe adverse reactions may occur even at non-toxic doses of vitamin D [25]. Therefore, it is imperative that clinicians prescribing vitamin D therapy are well-versed in appropriate dosing strategies and aware of potential side effects. In our study, no statistically significant differences were found among physicians regarding their knowledge of potential adverse effects associated with uncontrolled vitamin D administration.

Survey-based research is widely used in the medical field to evaluate various aspects, such as disease prevalence and patients' quality of life [26–28]. Additionally, survey studies are utilized to assess the adequacy of residency training

**Table 5. Comparison according to the frequency of rickets patient presentation.**

| Question | <1/week (n = 120) | ≥1/week (n = 78) | p |
|---|---|---|---|
| **What are the most common complaints of parents suggesting that their child has rickets?** | | | |
| Enlargement of the wrists and ankles | 8 (6.7) | 19 (24.4) | **<0.001** |
| Leg and knee deformities (genu varum, genu valgum) | 111 (92.5) | 77 (98.7) | 0.092 |
| Enlargement of the costochondral junction (rachitic rosary) | 10 (8.3) | 27 (34.6) | **<0.001** |
| Thoracal deformities | 7 (5.8) | 17 (21.8) | **0.001** |
| **What are the concerns that families convey to you about rickets?** | | | |
| Gait disorder | 56 (46.7) | 34 (43.6) | 0.671 |
| Decreased growth in height | 27 (22.5) | 30 (38.5) | **0.015** |
| Having bone fractures | 9 (7.5) | 47 (60.3) | **<0.001** |
| Deformity in bones | 87 (72.5) | 67 (85.9) | **0.027** |
| **What tests would you order for a child who is suspected of having rickets?** | | | |
| I'll do a referral to a pediatric endocrinologist without a test order. | 54 (45.0) | 24 (30.8) | 0.045 |
| Calcium | 56 (46.7) | 48 (61.5) | 0.041 |
| Phosphorus | 52 (43.3) | 25 (32.1) | 0.112 |
| Magnesium | 26 (21.7) | 11 (14.1) | 0.182 |
| Alkaline phosphatase | 47 (39.2) | 30 (38.5) | 0.921 |
| 25-OH-vit-D3 | 57 (47.5) | 54 (69.2) | 0.003 |
| Skeletal survey | 16 (13.3) | 12 (15.4) | 0.686 |
| Bone X-rays | 71 (59.2) | 51 (65.4) | 0.379 |
| **What kind of radiological findings would you expect to see in a child with a suspected case of rickets?** | | | |
| Metaphyseal enlargement, irregularity and clubbing | 108 (90.0) | 77 (98.7) | **0.016** |
| Growth plate enlargement | 63 (52.5) | 70 (89.7) | **<0.001** |
| Osteopenia in the long bones | 46 (38.3) | 51 (65.4) | **<0.001** |
| Stress fractures | 31 (25.8) | 56 (71.8) | **<0.001** |
| Pelvic deformities | 17 (14.2) | 43 (55.1) | **<0.001** |
| **Which of the following are types of rickets?** | | | |
| Nutritional (calciopenic, phosphopenic) | 99 (82.5) | 74 (94.9) | **0.010** |
| Vitamin D-resistant (familial hypophosphatemic) | 101 (84.2) | 78 (100) | **<0.001** |
| Vitamin D-dependent (genetic) | 90 (75.0) | 68 (87.2) | **0.037** |
| **Which option specifies the normal value of 25 OH vitD3?** | | | |
| <12 ng/ml <30 nmol/L | 4 (3.3) | 0 (0.0) | 0.155 |
| 12-20 ng/ml 30–50 nmol/L | 42 (35.0) | 21 (26.9) | 0.233 |
| >20 ng/ml >50 nmol/L | 69 (57.5) | 56 (71.8) | **0.042** |
| >100 ng/ml >250 nmol/L | 6 (5.0) | 4 (5.1) | >0.999 |
| **For your patients presenting with complaints/findings of rickets, would you question whether they are regularly taking the vitamin D treatment recommended by their Family Physicians under the age of 1 year?** | | | |
| Yes | 90 (75.0) | 69 (88.5) | **0.020** |
| **In patients with 25 OH-VitD3 deficiency, how would you proceed with conservative treatment?** | | | |
| I do not start treatment; I recommend consulting a pediatric endocrinologist | 94 (78.3) | 35 (44.9) | **<0.001** |
| I recommend a diet rich in calcium and phosphorus | 21 (17.5) | 42 (53.8) | **<0.001** |
| I start oral 25 OH vitD3 drops | 34 (28.3) | 14 (17.9) | 0.096 |
| I start oral 25 OH vitD3 ampoules | 3 (2.5) | 28 (35.9) | **<0.001** |
| I start intramuscular 25 OH vitD3 ampoules | 2 (1.7) | 19 (24.4) | **<0.001** |
| I start oral 25 OH vitD3 tablets/capsules | 3 (2.5) | 12 (15.4) | **0.001** |
| I recommend increasing the frequency of sunbathing | 42 (35.0) | 46 (59.0) | **0.001** |

*(Continued)*

**Table 5.** (Continued)

| Question | <1/week (n=120) | ≥1/week (n=78) | p |
|---|---|---|---|
| **What side-effect(s) are associated with vitamin D preparations if used in an uncontrolled way?** | | | |
| Weakness. fatigue | 63 (52.5) | 41 (52.6) | 0.993 |
| Anorexia | 33 (27.5) | 17 (21.8) | 0.367 |
| Bone pain | 51 (42.5) | 21 (26.9) | **0.026** |
| Kidney stone formation | 96 (80.0) | 70 (89.7) | 0.069 |
| Cardiac arrhythmia | 57 (47.5) | 21 (26.9) | **0.004** |
| Pancreatitis | 41 (34.2) | 23 (29.5) | 0.492 |
| Coma | 15 (15.0) | 4 (5.1) | **0.031** |

across specific specialties and subspecialties [29]. One of the most prevalent applications of such studies in the literature is the assessment of physician approaches to specific diseases. These evaluations facilitate interdisciplinary communication and help identify educational deficiencies or redundancies [30–33]. In the present study, we examined the clinical approaches of orthopedic surgeons, who are integral to the diagnosis and management of rickets, to elucidate their practices and perspectives regarding this condition.

To the best of our knowledge, this was the first national study in the English-language literature to investigate orthopedic surgeons' approaches to rickets patients within the context of pediatric endocrinology. A major strength of the study was the exclusive inclusion of board-certified orthopedic surgeons who participated voluntarily and without any conflicts of interest. Additionally, the high proportion of experienced physicians (nearly half with more than 10 years of practice and over 25% of pediatric patients in their daily caseload) contributed to the robustness of the findings. Moreover, the survey was distributed nationally, rather than being limited to a specific geographic region or institution, enhancing its generalizability.

However, several limitations should be acknowledged. First, the exact number of actively practicing orthopedic surgeons in the relevant email groups was unknown, and thus, the response rate to the survey could not be accurately calculated. Second, it is possible that the responses provided in the questionnaire did not fully reflect actual clinical practice, as discrepancies may exist between intended and real-world behavior. Nevertheless, previous research has shown that survey responses are generally consistent with daily clinical practices [34]. Third, this study was descriptive in nature and, therefore, did not allow for causal inference or evaluation of predictive relationships between variables. Fourth, we used self-reported data, which is subject to recall and self-report bias and may have affected our findings.

## Conclusions

Given the existence of various subtypes of rickets, timely consultation with a pediatric endocrinologist is essential to avoid inappropriate or inadequate management. This study highlighted significant differences in the clinical approach to rickets among orthopedic and traumatology specialists, influenced by factors, such as professional experience, institutional resources, and the availability of a pediatric endocrinologist. Newly diagnosed rickets cases evaluated in orthopedic and traumatology clinics should be referred to a pediatric endocrinologist to ensure accurate diagnosis and optimal treatment planning. Conversely, orthopedic surgeons with limited clinical experience may refer patients suspected of having rickets directly to a pediatric endocrinologist without performing any initial laboratory evaluations, resulting in unnecessary referrals. Therefore, it is critical that orthopedic clinicians perform at least basic diagnostic testing before initiating referral, to improve triage accuracy and resource utilization. To address these discrepancies and

enhance interdisciplinary collaboration, the development and integration of a standardized diagnostic and referral algorithm, based on current international guidelines, in orthopedic and traumatology residency training programs is strongly recommended.

The English in this document has been checked by at least two professional editors, both native speakers of English. For a certificate, please see: http://www.textcheck.com/certificate/jgKnx2

## Supporting information

**S1 Appendix.  Online survey applied to orthopaedic surgeons.**
(DOCX)

## Author contributions

**Data curation:** Niyazi Erdem Yaşar, Şenol Bekmez.

**Formal analysis:** Mehmet Ali Sungur, Şenol Bekmez.

**Investigation:** Banu Turhan, Mehmet Ali Sungur.

**Methodology:** Banu Turhan, Yalçın Turhan, Şenol Bekmez.

**Project administration:** Yalçın Turhan.

**Software:** Mehmet Ali Sungur, Yalçın Turhan.

**Supervision:** Banu Turhan, Niyazi Erdem Yaşar.

**Visualization:** Banu Turhan.

**Writing – original draft:** Banu Turhan.

**Writing – review & editing:** Niyazi Erdem Yaşar, Yalçın Turhan, Şenol Bekmez.

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
