## [Decision Letter · Decision Letter 0]

24 Jun 2025

Dear Dr. Turhan,

Thank you for submitting your manuscript to PLOS ONE. After careful consideration, we feel that it has merit but does not fully meet PLOS ONE’s publication criteria as it currently stands. Therefore, we invite you to submit a revised version of the manuscript that addresses the points raised during the review process.

We look forward to receiving your revised manuscript.

Kind regards,

Aamir Ijaz, MD, FCPS, FRCP, MCPS-HPE

Academic Editor

PLOS ONE

Journal Requirements:

2. We note that your Data Availability Statement is currently as follows: All relevant data are within the manuscript and in Supporting Information files.

Reviewers' comments:

Reviewer's Responses to Questions

**Comments to the Author**

1. Is the manuscript technically sound, and do the data support the conclusions?

Reviewer #1: Yes

Reviewer #2: Yes

2. Has the statistical analysis been performed appropriately and rigorously?

Reviewer #1: Yes

Reviewer #2: Yes

3. Have the authors made all data underlying the findings in their manuscript fully available?

Reviewer #1: Yes

Reviewer #2: Yes

4. Is the manuscript presented in an intelligible fashion and written in standard English?

Reviewer #1: Yes

Reviewer #2: Yes

Reviewer #1: This is a relevant and well-structured study that explores orthopedic surgeons' approaches to rickets management in Turkey, a topic with implications for interdisciplinary care and public health. The survey design is appropriate, and the manuscript effectively highlights practice variation and knowledge gaps. Few areas of improvemement that can be addressed include the following:

Could you elaborate on the methodology used to validate the survey instrument beyond expert review? Was there a pilot test or reliability assessment performed?

Were responses evenly distributed across regions and healthcare settings (public vs. private)? How might this affect generalizability?

How did you address the risk of response bias or social desirability bias in the self-reported data, particularly regarding knowledge and referral practices?

Reviewer #2: Please see highlighted text on the manuscript

Also! how does presence or absence of paed endocrinologist make any difference when it comes to complaints of patients OR concerns families convey about rickets?

Also on side-effects of Vit D preps?

**Do you want your identity to be public for this peer review?** For information about this choice, including consent withdrawal, please see our Privacy Policy

Reviewer #1: **Yes: ** Dr Sibtain Ahmed

Reviewer #2: **Yes: ** Muhammad Suhail Amin

---

## [Author Response · Author response to Decision Letter 1]

8 Jul 2025

Reviewer #1

Thank u for your valuable comments. We have attempted to revise the manuscript in accordance with your suggestions.

Q1: Could you elaborate on the methodology used to validate the survey instrument beyond expert review? Was there a pilot test or reliability assessment performed?

Yes, we conducted a pilot test of the survey instrument to validate its clarity, relevance, and consistency with 13 independent, board-certified orthopedic surgeons. Feedback from this phase was used to revise or clarify ambiguous items, ensure consistent interpretation of the questions, and exclude related data from the final analysis.

Q2: Were responses evenly distributed across regions and healthcare settings (public vs. private)? How might this affect generalizability?

How did you address the risk of response bias or social desirability bias in the self-reported data, particularly regarding knowledge and referral practices?

First, the study was conducted on a platform that all orthopedic surgeons across the country could access, and all orthopedic surgeons were reached and informed about the study; thus, an attempt was made to preserve the proportional distribution of education/research, government, and private health service providers across the country in the sample. Therefore, the sample is constituted of specialists from all over the country and is representative in this manner. However, the findings presented here only reflect the knowledge of Turkish orthopedic surgeons in Türkiye, which limits the generalizability of the results.

In terms of sample size, the data collection process continued until 10% (n=257) of the number of orthopedic surgeons (2500-2600) nationwide, which is reported in the literature to be sufficient as the rule of thumb (1,2), was reached, and 198 forms were included in the final analysis after excluding missing or inappropriate forms for analysis.

However, as you specified, we used self-reported data, which is subject to the risks of response or social desirability bias. This is a limitation in this study, as in every survey study, and the following sentence was added to the limitation section of “Discussion”.

“We used self-reported data, which is subject to recall and self-report bias and may have affected our findings.”

1. Krejcie RV, Morgan DW. Determining sample size for research activities. Educational and Psychological Measurement. 1970;30(3):607–10

2. Haberman SJ. Advanced statistics: A companion to statistical analysis of survey data. Springer; 1996.

Reviewer #2:

Thank u for your valuable comments. We have attempted to revise the manuscript in accordance with your suggestions.

Q1: Please see highlighted text on the manuscript

Line 173: The misspelled word “stuff” is replaced with “staff” and highlighted in the text.

Line 188: The "bone survey" is replaced with "skeletal survey" in all the text where it is mentioned and highlighted in the text.

Line 204: Table 4 titled "Comparison according to whether a pediatric endocrinologist is working in the institution" is discussed in detail in the following questions/answers:

Q2: Also! how does presence or absence of paed endocrinologist make any difference when it comes to complaints of patients OR concerns families convey about rickets?

We compared the presence or absence of a pediatric endocrinologist to determine whether there was heterogeneity in terms of complaints or concerns that families conveyed. We concluded that there was no significant difference (Table 4).

Q3: Also on side-effects of Vit D preps?

In general, all physicians were knowledgeable about not only common but also rare side effects.

Since multiple options could be selected in this survey question, no difference was observed between institutions with and without pediatric endocrinologists in their clinics regarding common side effects. However, a proportional difference was observed in rare side effects (cardiac arrhythmia and coma) due to the selection of multiple options. Since these results were considered to have no clinical significance, no emphasis was placed on them in the discussion.

We can attribute these meaningful responses to the two rare side effects, which may be due to the fact that orthopedic surgeons from institutions that employ pediatric endocrinologists participate in more meetings, such as education councils. Therefore, they could have a better understanding of rickets and its treatment. When we checked this statistically:

When institutions were compared in terms of the availability of pediatric endocrinologists, the proportions were as follows: 7.9% (3/38) in government hospital, 15.8% (3/19) in private hospitals, none in private practice, 92.8% (64/69) in education and research hospitals, 68.8% (22/32) in government universities, and 20.0% (4/16) in foundation/private universities. This difference was statistically significant (p<0.001).

When the institutions were grouped as the presence of a pediatric endocrinologist is typically expected (education and research hospitals, government and private universities) and considered collectively; the rate of presence of a pediatric endocrinologist was significantly higher (74.4%, 90/121) compared to other institutions (7.8%, 6/77) like government hospitals, private hospitals, and private practice (p<0.001).

As can be understood from our answer to Reviewer 1's second question (Q2);

“…an attempt was made to preserve the proportional distribution of training and research, public, and private health service providers across the country in the sample. Therefore, the sample is constituted of specialists from all over the country and is representative in this manner.”

---

## [Editor Report · Decision Letter 1]

24 Jul 2025

Management preferences of orthopedic surgeons in rickets patients in Turkey: results of a nationwide survey

PONE-D-25-19474R1

Dear Dr. Turhan,

We’re pleased to inform you that your manuscript has been judged scientifically suitable for publication and will be formally accepted for publication once it meets all outstanding technical requirements.

Kind regards,

Aamir Ijaz, MD, FCPS, FRCP, MCPS-HPE

Academic Editor

PLOS ONE
---

## [Editor Report · Acceptance letter]

PONE-D-25-19474R1

PLOS ONE

Dear Dr. Turhan,

I'm pleased to inform you that your manuscript has been deemed suitable for publication in PLOS ONE. Congratulations! Your manuscript is now being handed over to our production team.

Kind regards,

on behalf of

Professor Aamir Ijaz

Academic Editor

PLOS ONE